# Novel Epigenetic Biomarkers in Pregnancy-Related Disorders and Cancers

**DOI:** 10.3390/cells8111459

**Published:** 2019-11-18

**Authors:** Valentina Karin-Kujundzic, Ida Marija Sola, Nina Predavec, Anamarija Potkonjak, Ema Somen, Pavao Mioc, Alan Serman, Semir Vranic, Ljiljana Serman

**Affiliations:** 1Department of Biology, School of Medicine, University of Zagreb, 10000 Zagreb, Croatia; 2Centre of Excellence in Reproductive and Regenerative Medicine, University of Zagreb School of Medicine, 10000 Zagreb, Croatia; 3Department of Obstetrics and Gynecology, University Hospital Sestre Milosrdnice, 10000 Zagreb, Croatia; 4Department of Obstetrics and Gynecology, School of Medicine, University of Zagreb, 10000 Zagreb, Croatia; 5Clinic of Obstetrics and Gynecology, Clinical Hospital “Sveti Duh”, 10000 Zagreb, Croatia; 6College of Medicine, QU Health, Qatar University, Doha, Qatar

**Keywords:** cell-free DNA, cell-free fetal DNA, cell-free tumor DNA, extracellular vesicles, exosomes, microRNA

## Abstract

As the majority of cancers and gestational diseases are prognostically stage- and grade-dependent, the ultimate goal of ongoing studies in precision medicine is to provide early and timely diagnosis of such disorders. These studies have enabled the development of various new diagnostic biomarkers, such as free circulating nucleic acids, and detection of their epigenetic changes. Recently, extracellular vesicles including exosomes, microvesicles, oncosomes, and apoptotic bodies have been recognized as powerful diagnostic tools. Extracellular vesicles carry specific proteins, lipids, DNAs, mRNAs, and miRNAs of the cells that produced them, thus reflecting the function of these cells. It is believed that exosomes, in particular, may be the optimal biomarkers of pathological pregnancies and cancers, especially those that are frequently diagnosed at an advanced stage, such as ovarian cancer. In the present review, we survey and critically appraise novel epigenetic biomarkers related to free circulating nucleic acids and extracellular vesicles, focusing especially on their status in trophoblasts (pregnancy) and neoplastic cells (cancers).

## 1. Introduction

It has been long known that many diseases categorized in the same group, such as cancer, are genetically quite heterogeneous. The method that led to this realization is based on the discovery of circulating cell-free DNA (cfDNA) in 1948, when Mandel and Metais identified its presence in the plasma and serum of both healthy and sick individuals [1]. However, the significance and role of cfDNA was elucidated only in the second part of the 20th century [2]. In 1977, Leon et al. found higher concentrations of cfDNA in the sera of cancer patients in comparison with healthy population, which suggested that cfDNA could have an oncologic role [3]. More than a decade later, Stroun et al. demonstrated that cfDNA found in cancer patients harbored fragments from cancer cells [4]. This was followed by the discovery of *RAS* point mutations in the fragments of cfDNA originating from cancer cells [5,6], which marked the beginning of liquid biopsy profiling as a diagnostic method and brought cfDNA into the focus of research interests.

Liquid biopsy is a minimally invasive method for the detection and quantification of genetically important alterations within the cfDNA [7] (Figure 1). It is faster and more efficient than classic biopsy and, therefore, can be used repetitively. For a successful clinical application of liquid biopsy, it is crucial to standardize analytical methods and pre-analytical procedures, including plasma separation and selection of the optimal isolation assay, that may yield a sufficient amount of high-quality DNA. Multiple studies confirmed that blood sampling and processing might significantly affect DNA yield and downstream analyses [8]. However, despite the substantial efforts to standardize and optimize the methodology, such as those of the European FP7 consortium SPIDIA4P (standardization and improvement of generic pre-analytical tools and procedures for in-vitro diagnostics, http://www.spidia.eu/) [9], no consensus has been reached on the pre-clinical preparations for liquid biopsy [10].

Aberrant DNA methylation can be detected in different pathological conditions. It was first observed some 40 years ago when a global methylation analysis by chromatographic methods revealed significantly reduced DNA methylation levels in different types of malignancies compared with normal tissues [11,12,13]. Since gene expression can be inhibited by DNA methylation, it was realized that the inactivation of tumor suppressor genes is a fundamental process in oncogenic transformation. Consequently, many studies investigated aberrant epigenetic mechanisms in various cancer subtypes [14]. These alterations have been detected in the cfDNA of cancer patients, indicating the great potential of aberrant DNA methylation as a diagnostic biomarker in cancer detection [15].

Circulating cell-free fetal DNA (cffDNA) was discovered in 1997 [16] and only three years later, it was possible to extract it from mothers’ blood cells [17]. Higher concentrations of cffDNA in the blood of a pregnant woman carrying a child with trisomy 21 (Down syndrome, OMIM#190685), compared with pregnant women carrying a healthy child, opened a new avenue to non-invasive prenatal testing [18]. Today, cffDNA is widely used in aneuploidy screening, but it is still not used in the clinical evaluation of pregnancies complicated by disorders, such as pre-eclampsia (PE) [19,20,21] or intrauterine growth restriction (IUGR), although several studies showed that cffDNA levels were increased in these pathological conditions [22,23,24].

Besides cfDNA, human plasma and serum contain various classes of RNA molecules, including protein-coding messenger RNAs (mRNAs); small non-coding RNAs (sncRNAs), such as microRNAs (miRNAs), piwi-interacting RNAs (piRNAs), transfer RNAs (tRNAs), small nucleolar RNAs (snoRNAs), small nuclear RNAs (snRNAs), and miscellaneous RNAs (misc-RNAs); and long non-coding RNAs (lncRNAs) [25]. These circulating RNAs also have the potential to serve as biomarkers.

Circulating RNAs and cfDNA are usually packed in extracellular vesicles (EVs) [25,26], another promising tool for early diagnosis detectable with liquid biopsy.

EVs are membranous particles released by a variety of cells into the extracellular space. They are involved in intercellular communication, transferring the information from donor to recipient cell independent of direct cell–cell contact. Based on their biogenesis and size, EVs are subdivided into four subclasses: oncosomes, apoptotic bodies, microvesicles, and exosomes [27,28]. These vesicles contain proteins, lipids, and nucleic acids (DNA and various classes of RNA molecules) specific for their cells of origin, thus serving as indicators of cell state. They may be found in body fluids, including blood, plasma, urine, saliva, amniotic fluid, breast milk, and pleural effusion [29,30]. Since EVs are implicated in various physiological and pathophysiological conditions, they could have a great potential as a disease detection tool, especially exosomal miRNAs, the best described class of sncRNAs.

We intend to explore the role of these molecules as potentially powerful and useful tools in the diagnosis, treatment, and follow-up of malignant and pregnancy-related diseases from the epigenetic point of view. This review will encompass two seemingly disparate, but genetically and epigenetically similar, tissues—placenta and cancer. For the placenta to fulfill adequately its function of supporting the growth of an embryo and fetus, trophoblasts must properly invade the endometrium and remodel the mother’s spiral arteries. This process resembles the neoplastic process. Thus, it is not surprising that placentation and tumorigenesis share some common physiological characteristics, which can be observed at the genetic and epigenetic levels [31,32].

The most important difference between placentation and tumorigenesis is that the invasion of trophoblasts is spatially and temporally regulated; therefore, the invasion is neither too shallow, as it is in PE and IUGR, nor too deep, as it is in placenta accreta or percreta. Previous studies emphasized the similarities between placental and tumor tissues and research mostly progressed in the direction of using the knowledge about trophoblast behavior to better understand tumor behavior [33,34]. In recent years, there have been more and more studies where a certain molecule found to be important for the behavior or targeting of tumor cells has subsequently been associated with the functioning of trophoblasts and gestational diseases [35,36,37,38,39,40,41,42,43,44,45,46,47,48,49,50,51]. To bring the reader closer to the premise that the same molecules can mediate tumor and trophoblast behavior, we will discuss some of these studies in the paragraph below.

The overexpression of enhancer of zeste homolog 2 (EZH2) protein (histone methyltransferase) is associated with the aggressiveness of some cancers. In the placenta, it induces trophoblastic invasion and is found at reduced levels in placentas from recurrent miscarriages [37]. The chromatin-remodeling proteins, histone deacetylases (HDACs), are greatly increased in various human cancers and promote their progression. One such deacetylase, HDAC9, promotes the migration and invasion of trophoblast cells by inhibiting the expression of *TIMP3* (tissue inhibitor of metalloproteinases 3) via hypoacetylation of its promoter histone. In pre-eclamptic placentas, the expression of HDAC9 was also reduced [38].

Uridine diphosphate (UDP)-GalNAc transferase 3 (GALNT3), which initiates the *O*-GalNAc glycosylation of many proteins (including E-cadherin) in the Golgi apparatus, has a conserved function in trophoblast stem cells, blastocysts, trophoblast, and human mammary epithelial cells. In human mammary epithelial cells, GALNT3 is considered to maintain an epithelial phenotype since the loss of its expression reduces *O*-GalNAc glycosylation and E-cadherin expression, induces epithelial–mesenchymal transition (EMT), and is associated with more aggressive, mesenchymal (claudin-low) type of breast carcinomas [39].

Let-7 miRNAs inhibit proliferation and invasion in many cancers (colon, lung, hepatocellular, gastric, pancreatic, ovarian, prostate, and melanoma cancers) and have the same effect on trophoblast cells. A reduction of Let-7 expression leads to increased proliferation, migration, and invasion of extravillous trophoblasts through upregulation of *MDM4* (murine double minute 4), which acts as a p53 protein inactivator. In pre-eclamptic placentas, Let-7 is reduced, as well as MMP-2 and MMP-9 metalloproteinases, and their inhibitors are increased [40]. In contrast to Let-7 miRNAs, miRNA-34a-5p reduces the migration and invasion of trophoblasts (via Smad4, which is associated with invasiveness and metastasis), whereas a reduction in miRNA-34a-5p stimulates trophoblast invasion and migration (by raising Smad4 levels) [41].

What is important is not only which tissues or cells invade target tissues, but also where this invasion takes place and what kind of communication exists between the invasive and target tissues [42,43]. Some investigators “blame” cancers for acquiring the phenotypic features of a trophoblast, such as autocrine growth, metabolism through adaptive enzymatic pathways, ability to survive under inadequate conditions, anti-apoptotic factors, latent mesenchymal traits, amoeboid motion or transfer through fetomaternal microchimerism, changes in cadherins, creation of giant cells, acytokinetic DNA replication, changes in the host (such as in angiogenesis), immunological tolerance, and β-hCG (β-human chorionic gonadotropin) secretion [44].

We must, however, be aware of the many differences between trophoblasts and cancer cells in the acquisition of migratory and invasive phenotypes (described in detail in [45,46,47]). The pivotal role in successful placentation is played by the cell column of anchoring villus, where the epithelial cells of the trophoblasts (located in the proximal chorion) differentiate into distal invasive and migratory cells of the mesenchymal phenotype [48].

The methylome of extravillous trophoblasts (EVTs; composed of invasive and migratory trophoblast cells) differs significantly from the methylome of cytotrophoblasts (cells of epithelial phenotype) because EVT genes involved in cell adhesion and proliferation are downregulated. Of the 102 genes that are differentially expressed between invasive EVTs and cytotrophoblasts, only 41 are associated with EMT and cancer metastasis, which suggests that trophoblasts retains their unique invasive phenotype that tumors are unable to replicate in full [49]. This may also be due to the fact that a healthy trophoblast is constantly in contact with the environment it invades and adapts to [50], whereas a tumor is an indifferent imitator. Thus, according to their behavior, trophoblast invasion leads to the birth of a new life, whereas cancer invasion in many cases has the opposite result.

We need to keep in mind that cellular epigenetic patterns found at the time of tissue collection of either a trophoblast or cancer reflects not only the present state, but also the memory of the past and instruction for the future—it could be said that the epigenome carries information on the past, present, and future [51]. Therefore, monitoring the epigenetic changes in extracellular DNA, as free DNA or bound in EVs, and monitoring circulating RNA are essential for observing the state of health and disease.

## 2. Circulating Cell-Free DNA

Circulating cell-free DNA (cfDNA) is a short fragment of double-stranded DNA present in blood and other body fluids [3,52,53,54]. Most cfDNA molecules are 150-bp long, which corresponds to the main DNA organization unit within the nucleosome and supposedly, results from the activation of nuclease during apoptosis [55]. Due to their electrostatic and autocondensation properties, DNA molecules can create compounds with other molecules and structures [56]. Therefore, they can be released in the blood as molecules or macromolecular complexes [57,58] that protect the DNA molecules against nucleases and the immune system [59]. 

The origin and modes of transfer of cfDNA are still poorly understood. In healthy individuals, plasma cfDNA originates mostly from hematopoietic cells (55% from white blood cells and 30% from erythrocyte progenitors), whereas 10% of plasma cfDNA comes from vascular endothelial cells and 1% from hepatocytes [60]. It is thought to be related to and maintained by a variety of cellular processes, such as apoptosis, necrosis, phagocytosis, and active secretion [61]. More than 93% of amplifiable cfDNA in plasma is located in plasma exosomes, which suggests that it is actively released from the cells [26]. We should differentiate between disease-related and normal cfDNA, because cfDNA can be isolated not only from healthy individuals, but also from individuals suffering from inflammatory diseases [62], trauma [17,63] or cancer.

Although cfDNA can be considered a transient passenger, it has been suggested to play an active role in pathological and physiological conditions [64]. The active release of DNA molecules is very important in the maintenance of cell homeostasis. A damaged DNA, such as an oxidized mitochondrial DNA (mtDNA), is harmful to cells, so its removal by secretion through exosomes is crucial for the maintenance of mitochondrial homeostasis [65]. The reason why living cells secrete normal, undamaged DNA is unknown. In physiological conditions, cfDNA released by neutrophil extracellular trap (NETosis) may be involved in antimicrobial immune responses, while in pathological conditions, it may promote cancer growth and establishment of a metastatic niche or development of thrombotic complications and vascular dysfunction in cancer patients [64,66,67]. cfDNA may also damage the DNA of healthy cells by integrating into their genomes; it may also influence the function of recipient cells via extracellular vesicle-mediated transfer and mediate communication between cells [68,69].

### 2.1. Circulating Cell-Free Tumor DNA

Particularly interesting is the cell-free tumor DNA (ctDNA), a fraction of cfDNA released into the circulation as a result of cancer cell death (apoptosis and/or necrosis) [70] or active secretion of EVs, such as exosomes and prostasomes (prostate-specific EVs) [71]. ctDNA can also be detected in other body fluids, such as urine [72], stool [73], saliva [74], pleural fluid [75], and cerebrospinal fluid [76].

Cancer is one of the leading causes of death in the world, so there is an unmet and urgent need to optimize biomarkers of early cancer detection. Since some cancers, such as lung cancer, ovarian cancer, and non-Hodgkin lymphoma, are often detected at an advanced stage, diagnostic assays that could enable their early detection are under intense investigation and development [77]. For example, although ovarian cancer accounts for only 2.5% of all female malignancies, it is responsible for 5% of all female cancer-related deaths. The low survival rates are caused by the late detection (advanced stage) [78]. The improvement of prevention and early detection is a research priority because ovarian cancer diagnosed at an early stage has a 5-year survival rate of >90% [78].

Tissue biopsy is still considered the gold standard for the diagnosis of cancer [79]. Its disadvantages include invasiveness of the procedure [80], a small sample that cannot reflect the cancer heterogeneity (morphologic, molecular, and epigenetic) [81], molecular discrepancies (genetic, epigenetic, and transcriptomic) between the primary and metastatic cancer, and the fact that most studies usually carry out these evaluations only once [15]. All these factors have created a need for new diagnostic methods. In addition, a classic biopsy may be challenging to perform in cancers that are anatomically inaccessible for sampling, such as pancreatic or brain cancer [82].

Improving cancer prevention and early detection presumes the existence of specific diagnostic biomarkers that could reliably detect cancers in their early phase [83]. One such biomarker is ctDNA, because it is specific for cancer cells. As the primary purpose of detection and quantification methods is the assessment of clinically important cancer alterations, the methods that are used must be highly sensitive for ctDNA and able to cover a large spectrum of cancer aberrations. Preferences should be given to methods characterized by simple sampling, minimal invasiveness and morbidity, and low cost, but still allowing for the assessment of tumor heterogeneity [84]. Assays that are currently used to detect and measure ctDNA include polymerase chain reaction (PCR) and next-generation sequencing (NGS) methods. PCR-based assays have a faster turnaround time and are much cheaper, but can only assess one or a few specific mutations at a time. In contrast, an NGS assay allows for the assessment of a broader range of genomic alterations because sequencing can detect alterations anywhere within the captured regions (source: https://www.cap.org/member-resources/articles/the-liquid-biopsy, visited 6 August 2019). Of note, liquid biopsy assays have already been utilized in clinical practice (e.g., Trovera™ EGFR urine-based liquid biopsy (three genes are covered: *EGFR*, *KRAS*, and *BRAF*) and Cobas® EGFR mutation test v2 that covers the *EGFR* gene)) [8]. However, many issues related to the routinized use of liquid biopsy remain to be resolved, including pre-analytical sample handling, appropriate assays for cfDNA extraction, and optimal workflow for analyses [10].

The early detection of relapse (recurrence) is currently based on the presence of clinical symptoms, blood biomarkers with various sensitivity and specificity (e.g., carcinoembryonic antigen (CEA) for colorectal carcinoma (CRC), prostate-specific antigen (PSA) for prostate cancer, and cancer antigen 15-3 (CA 15-3) for breast cancer), and radiologic evidence of the disease [85]. No effective assays are available to accurately predict disease recurrence. A liquid biopsy for ctDNA detection could be an efficient approach in the follow-up of cancer patients (Figure 1). Tie et al. analyzed ctDNA in patients with stage II CRC after surgical resection. The study demonstrated that the presence of ctDNA after the treatment indicated the existence of residual cancer, implying the link between ctDNA and high risk of disease relapse. Equally, ctDNA had a better predictive value than the conventionally used CEA assay [86]. Other studies also confirmed the effectiveness of ctDNA in the early diagnosis of relapse in breast cancer [87], lung cancer [88], gastric cancer [89], pancreatic cancer [90], and diffuse large B-cell lymphoma (DLBCL) [91]. All the aforementioned studies pointed out the importance of ctDNA in the detection of relapse several months before the standard methods.

A significant decline of ctDNA level shows a positive correlation with treatment effects [73], while the appearance of ctDNA or a significant increase in ctDNA during the treatment indicates the progression or resistance of the disease to therapy [92]. For example, if the cfDNA level remains high in patients undergoing platinum-based chemotherapy for ovarian cancer, it might indicate the presence of residual cancer cells [93]. The levels of cfDNA in breast cancer patients with localized disease can decrease after surgery to levels observed in healthy individuals [94]. Bettegowda et al. used digital PCR to analyze ctDNA in 640 cancer patients at different stages of the disease and found that 47% of cancers in the early stage and 55% of those in the advanced stage could be detected with ctDNA and that the level of ctDNA depended on cancer origin [95]. 

With ctDNA analyses, new cancer mutations can be also detected, including those that confer therapeutic resistance. A study of Gormally et al. carried out on 1000 healthy volunteers determined the presence of *KRAS* (Kirsten rat sarcoma 2 viral oncogene homolog) and *TP53* (tumor protein p53) mutations in 33 volunteers, and 16 of them developed cancer in the mean period of 18.3 months [96]. The presence of *KRAS* mutations in plasma may be a marker of poor response to chemotherapy [97]. In 2016, the Food and Drug Administration (FDA) approved the Cobas® EGFR mutation test v2 (Roche) for blood and tissue analysis of *EGFR* mutations in patients with non-small-cell lung cancer (NSCLC). More than 50% of NSCLC patients treated with tyrosine kinase inhibitors will develop secondary *EGFR* mutations, resulting in resistance to this therapy [98] and consequent use of osimertinib, a drug used for the treatment of NSCLC with *EGFR* mutations. A ctDNA analysis may be helpful in the detection of *EGFR* T790M mutation in NSCLC patients and prediction of outcome to osimertinib therapy [99].

On the basis of these studies, we believe that a proportion of human cancers can be detected by ctDNA analyses, although the presence of a gene mutation does not necessarily indicate the presence of cancer. Mutations in healthy individuals can originate from mutations in hematopoietic cells [100]. The rate of false-positive results could be diminished by a broad sequencing cfDNA that enables the detection of multiple mutations related to cancers [88].

In healthy individuals, the total cfDNA concentration is about 30 ng/mL of plasma (range: 0–100 ng/mL), whereas in those with cancer, it can be as high as 1000 ng/mL due to insufficient clearance processes [101,102]. The proportion of cfDNA that originates from cancer cells varies depending on the state and size of the cancer. The amount of cfDNA is also influenced by clearance, degradation, and other physiological filtering processes in the blood and lymphatic circulation [15]. The main organs involved in the elimination of cfDNA are liver, spleen, and kidneys [64]. It is known that cfDNA is cleared from the circulation rapidly and efficiently regardless of its size or configuration [103]. For example, ctDNA has a high elimination rate in plasma (its half-life ranges from 16 min to 2.5 h) [73], which makes it a suitable dynamic biomarker for tracking cancer biology and progression and response to therapy [104,105].

The fraction of ctDNA in the total cfDNA is very low (~1%), which poses a major problem in its isolation and diagnostics [105,106,107]. For optimal use of ctDNA as a diagnostic tool, it is important to define its biological characteristics, such as its existing form (nucleosome, single-stranded, or double-stranded), size (the size of ctDNA is still undetermined), mode of entrance into the bloodstream (it may originate from apoptotic or necrotic tumor cells, living tumor cells, or circulating tumor cells, or it may have multiple origins), and biological function (potential metastatic component) [108]. It is also important to develop a method that is sensitive enough to detect low plasma concentrations of ctDNA, and define its analytical and clinical validity [108]. Before cfDNA may be used in clinical practice, there are several challenges that need to be overcome: technological and analytical limitations, sampling noise, diagnostic specificity and sensitivity, and ambiguity related to cancer-associated mutations (described in detail in [109]).

### 2.2. Epigenetic Alterations in ctDNA

Epigenetic mechanisms include DNA methylation, histone modification, and chromatin remodeling. These processes are closely associated with gene regulation [110,111]. DNA methylation is a process of conversion in a DNA sequence, catalyzed by DNA methyltransferase (DNMT), where the methyl group is added to cytosine resulting in 5-methylcytosine (5mC) formation, which usually lies next to a guanine base (CpG methylation). Many cellular processes, such as genomic imprinting, chromosome stability, chromatin structure, development and cellular differentiation, X-chromosome inactivation, and transcription, are regulated by DNA methylation [112]. More than three decades ago, cancer-specific aberrant DNA methylation patterns were first detected in cancer tissues [113]. In normal cells, it is crucial to maintain the balance of DNA methylation between proto-oncogenes and tumor suppressor genes. However, if the promoter regions (CpG islands) of these tumor suppressor genes undergo hypermethylation, the genes will be silenced, causing the cells to escape from the normal cell cycle and take a path toward tumorigenic transformation [114]. Another characteristic of cancer cells is global hypomethylation of DNA. This process promotes the transcription of genes, including proto-oncogenes, which contribute to cancer formation [115].

For tumorigenesis, dysregulation of specific epigenetic mechanisms plays an almost as important role as genetic alterations [111]. Therefore, much effort has been made to identify CpG island hypermethylation in certain genes that could be used as a cancer biomarker [14] (Table 1). As the stability of the information provided is a prerequisite for clinical usefulness of biomarkers, DNA methylation as a stable modification in cancer cells has become a promising candidate as a potential biomarker for early cancer detection [116]. DNA methylation of certain sets of genes principally reflects the typical clinicopathological features, such as disease subtype, patient outcome, and cancer therapy response [14]. In addition, a DNA methylation analysis enables the detection of the origin of cfDNA and the cancer site [117] in addition to early detection of primary cancer and metastases to other organs [118]. 

Moreover, specific gene methylation detected in circulating DNA before treatment can be a predictive biomarker for anticancer therapy efficacy and disease prognosis. For example, methylation of *14-3-3 sigma,* a major G2-M checkpoint control gene for many of the protein kinases and phosphatases, could be a predictor of longer survival for NSCLC patients receiving platinum-based chemotherapy [131].

The use of methylated cfDNA for cancer detection is rapidly gaining attention. Shen et al. demonstrated a robust performance of plasma cfDNA methylomes in cancer detection and classification [132]. Up to date, the only officially FDA-approved gene test related to ctDNA methylation is the one for CRC detection: *SEPT9* (septin-9). *SEPT9* plays an important role in cell mitosis and cell cycle regulation and acts as a tumor suppressor [119]. Consequently, *SEPT9*-reinforced promoter DNA methylation in ctDNA is a significant predictor of premalignant and malignant changes in colorectal carcinogenesis [120]. Apart from *SEPT9*, there are other genes, such as *RASSF1* (Ras association (RaIGDS7AF-6) domain family number 1) and *BRCA1* (breast cancer 1), whose methylation changes could indicate malignant transformation. Hypermethylated *RASSF1* and *BRCA1* were found in the plasma of ovarian cancer patients [123,133]. Furthermore, recent studies with methylated *WIF1* (Wnt inhibitory factor 1) and *NPY* (neuropeptide Y) showed that hypermethylation of their gene promoters is detectable in both CRC tissue and corresponding ctDNA [121], while post-surgery detection of methylated *BCAT1* (branched-chain amino acid transaminase 1) and *IKZF1* (IKAROS family zinc finger 1) in ctDNA in CRC patients was associated with increased risk of residual disease and its subsequent recurrence [122]. *WNT5A* (Wnt family member 5A)*, KLK10* (kallikrein-10)*, MSH2* (mutS protein homolog 2)*,* and *SOX17* (SRY-box 17) were found to be more frequently methylated in ctDNA of breast cancer patients than healthy individuals [124]. More intense methylation of *WNT5A* correlated with a greater tumor size, advanced disease, and poor outcome; *SOX17* methylation significantly correlated with the incidence of death, shorter progression-free survival, and overall survival; and *KLK10* methylation significantly correlated with unfavorable clinicopathological characteristics and relapse [124]. Promoter methylation of six genes (*APC*, *CCND2*, *FOXA1*, *PSAT1*, *RASSF1A*, and *SCGB3A1*) in ctDNA in breast cancer patients was confirmed as cancer-specific, although with a variable performance. A panel combining four of them (*APC*, *FOXA1*, *RASSF1A*, and *SCGB3A1*) revealed the highest accuracy for disease detection (94%) [125]. In primary NSCLC and serum ctDNA, promoter methylation of a panel of six genes (*CDO1*, *HOXA9*, *AJAP1*, *PTGDR*, *UNCX*, and *MARCH11*) was found to be potentially useful as a detection and prognostic biomarker [126]. The methylation levels of eight genes (*CALCA*, *HOXA9*, *RASSF1A*, *CDKN2A*, *DLEC1*, *CDH13*, *PITX2*, and *WT1*) in ctDNA were significantly higher in lung cancer patients than in non-lung cancer patients. Three of them (*RASSF1A*, *CDKN2A*, and *DLEC1*) were found only in lung cancer patients [127]. There were also other genes whose methylation was associated with lung cancer. A genome-wide plasma cfDNA methylation profiling identified ten genes (*B3GAT2*, *BCAR1*, *HLF*, *HOPX*, *HOXD11*, *MIR1203*, *MYL9*, *SLC9A3R2*, *SYT5*, and *VTRNA1-3*) as potential biomarkers for this disease [128]. A recent study of Wasenang et al. revealed that cfDNA methylation of *OPCML* (opioid binding protein/cell adhesion molecule-like) and *HOXD9* (homeobox D9) could serve as a differential biomarker for cholangiocarcinoma and other biliary diseases [130]. The success of therapy and follow-up of prostate cancer patients is traditionally evaluated by PSA levels and bone scintigraphy findings, but now it is shifting toward the measurement of blood biomarkers of DNA methylation in *GSTP1* (glutathione S-transferase pi 1) and *APC* (adenomatous polyposis coli), both of which have a prognostic role [129]. Apart from 5mC, there are also 5-hydroxymethylcytosine (5hmC) signatures in ctDNA. 5-Hydroxymethylome in cfDNA may serve as a potential biomarker for NSCLC patients [134].

The largest study up to date that analyzed DNA methylation in association with epithelial ovarian carcinoma (EOC) risk was conducted on ~20,000 ovarian carcinomas that were previously included in a genome-wide association study (GWAS) of ovarian cancer. The study identified five genes (*MAPT*, *HOXB3*, *ABHD8*, *ARHGAP27*, and *SKAP1*) that exhibited significant changes in their DNA methylation profile compared with healthy controls. The authors proposed that regulating the expression of these genes through methylation might affect EOC risk [135]. In neuroendocrine tumors of the pituitary gland, *RASSF1* methylation correlated with less invasive tumors [136]. Our previous study on high-grade serous ovarian carcinomas revealed *SFRP1* (secreted frizzled-related protein 1) as an important predictor of malignant transformation. The loss of SFRP1 protein in ovarian carcinomas was attributed to the hypermethylation of its gene promoter region [137]. Therefore, in our opinion, the ultimate goal is to identify a highly relevant methylation gene (HRMG) panel, such as *RASSF1* and *SFRP1*, whose analysis of ctDNA would enable earlier detection of various cancers [138]. Kel et al. introduced a new term “epigenomic walking”, suggesting the alterations of genome methylation in cancer that lead to the methylation of regulatory regions of important genes. They defined a group of six genes (*CALC*, *ENO1*, *MYC*, *PDX1*, *TCF7*, *ZNF43*) whose methylation was strongly related to the development of CRC [139]. It would be interesting to examine the DNA methylation of the above-mentioned genes in the blood of cancer patients.

Another potentially powerful tool whose epigenetic modification could help in early cancer detection are changes in histone methylation. Histones are relatively stable in circulation and are released from the cell with DNA molecule fragments [140]. Since histone deacetylase and histone methyltransferase inhibitors monitor changes in circulating histones and DNA methylation patterns, their possible antitumor effects and role in improved cancer screening should be assessed [141,142].

A study of Rahier et al. revealed that CRC patients had increased concentrations of cell-free nucleosomes (cf-nucleosomes) in the blood, and the panel that analyses 12 epigenetic nucleosomal epitopes seemed to be a promising approach in the identification of individuals with a higher risk for CRC [143]. In addition, Gezer et al. found that CRC patients had lower concentrations of H3K27me3 (trimethylations of lysine 9 on histone 3) and H4K20me3 (trimethylations of lysine 20 on histone 4). The authors concluded that the combination of these two biomarkers could be a new diagnostic tool in the detection of CRC [144]. This approach could be promising in the detection of other cancer types that are difficult to detect in the early phase when the treatment success rate would be much higher (e.g., pancreatic cancer where epigenetic changes in cf-nucleosomes in the blood were also previously reported) [145].

### 2.3. Circulating Cell-Free Fetal DNA

In 1997, Lo et al. measured cffDNA as a marker of the Y chromosome in order to determine fetal sex [16]. Although the main source of cffDNA is the placenta [146], this name has been proposed because shortly after delivery, fetal DNA cannot be detected in mothers’ plasma anymore, while fetal cells can be detected even months after delivery [147]. Equally, in placental mosaicism, fetal DNA shows the characteristics of the placenta rather than the fetus [148]. Currently, previously mentioned *RASSF1* gene hypermethylation is used for the distinction between maternal hypomethylated DNA and circulating fetal hypermethylated DNA in maternal blood [149].

The main source of cffDNA is the villous trophoblast. During pregnancy, these cells undergo a physiological cycle that is crucial for the normal function of the placenta and starts with the proliferation and differentiation of placental stem cells. A subsequent process is the fusion of the differentiated cytotrophoblasts and formation of syncytiotrophoblasts. After that, syncytiotrophoblasts undergo apoptosis and will be released into the maternal circulation in the form of syncytial nodules [150]. Given that the release of the syncytial nodules can also be caused by aponecrosis and reinforced apoptosis in pathological conditions, such as PE and IUGR, increased levels of cffDNA in the maternal blood can be ascribed to the same processes [151].

The percentage of placental fraction in maternal circulation is crucial for obtaining informative and reliable results [152]. Fetal fraction is the ratio between cffDNA and overall cfDNA levels (including both fetal and maternal cfDNA) [153]. While fetal cfDNA originates from placenta, maternal cfDNA originates from the apoptosis of hematopoietic cells [154]. In addition, lower fetal fraction can be the result of increased level of maternal cfDNA [155]. A poor accuracy of the test is related to low fetal fraction [156]. Consequently, most manufacturers avoid presenting their test results if placental fraction is lower than 4% [152].

The fetal fraction has a bell-like shape distribution that reaches its maximum of 10–20% between the 10th and 21st week of gestation [157]. It increases with gestational age [155], fetal crown–rump length (CRL) [157], and serum level of free β-hCG [157,158]. It is also higher in smokers, pregnancies affected by trisomy 21 (Down syndrome) [157], and a natural conception compared with in vitro fertilization (IVF) [158]. On the other hand, factors linked with lower fetal fraction include increased maternal body mass index (BMI) [155,157,158], advanced age [155,158], IVF conception, multiparity, chronic hypertension, and diabetes mellitus [155]. Notably, the size of the placenta does not affect the level of fetal fraction [159]

Not only is the genetic information obtained from the fetus the basis for the identification of genetic abnormalities, but it also underscores some pathologies associated with the mother [160]. The detection of cffDNA opened a new venue for non-invasive prenatal testing (NIPT), which is now used for the early detection of various disorders, including aneuploidies, single-gene disorders (e.g., cystic fibrosis, Huntington’s disease, thanatophoric dysplasia, and achondroplasia) [161,162,163,164,165,166], and X-linked genetic disorders (e.g., Duchenne muscular dystrophy) [163,167]. NIPT can be performed from gestation week 9, i.e., earlier than invasive testing, which cannot be performed safely before week 11 [168,169], and much earlier than an ultrasound anomaly scan. It is more accurate than conventional combined screening (nuchal translucency, pregnancy-associated plasma protein-A (PAPP-A), and hCG) in the first trimester [170]. In addition, it is non-invasive and avoids the ~1% risk of miscarriage associated with amniocentesis/chorionic villus sampling procedures [169]. It also allows for a timely therapeutic intervention in conditions such as congenital adrenal hyperplasia (CAH) [171].

As already mentioned, cffDNA can be also used in the detection of some gestation-related disorders, such as PE and IUGR. IUGR is a pregnancy condition with various etiology and characterized by the failure of the fetus to achieve its normal growth potential. It is associated with a substantial perinatal morbidity and mortality and risk for cardiovascular diseases in adult life [172]. A number of different causes have been attributed to the development of IUGR, but placental dysfunction is thought to play a key role. Studies found cffDNA levels to be significantly higher in IUGR-complicated pregnancies than in normal pregnancies [22,23,24]. Moreover, concentrations of cffDNA were found to be significantly higher in pregnancies complicated with hyperemesis gravidarum (HG) compared with normal pregnancies [173].

PE is a gestational disease that affects 2–8% of all pregnancies [174]. During the normal process of placentation, invasive endovascular trophoblasts invade maternal spiral uterine artery walls and transform them to low-resistant, high-throughput blood vessels [175]. In PE, because of a shallow and incomplete invasion, spiral arteries are not transformed, which leads to inappropriate placental perfusion and subsequent hypoxia [176]. Because of the reinforced trophoblasts, apoptosis and aponecrosis as a consequence of ischemia, inflammatory response, and reduced elimination of cffDNA from maternal circulation, higher concentrations of cffDNA were detectable in maternal circulation [177]. So far, numerous studies have explored elevated levels of cffDNA in pregnant women before the onset of PE, as well as in those with already diagnosed PE [19,20,21,178]. Before the onset of PE, cffDNA levels were elevated in two cases: first time, between the 17th and 28th week of gestation and second time, three weeks before the onset of symptoms [179].

Rolnik et al. found that the first trimester markers for PE, mean arterial pressure (MAP) and uterine artery pulsatility index (UtAPI), showed a negative association with fetal fraction, while PAPP-A and placental growth factor (PIGF) had a significantly positive association with fetal fraction [155]. However, as the analyzed levels of cffDNA in the first trimester of gestation varied markedly, their association with PE has not been confirmed [180,181,182]. Yu et al. prospectively analyzed the levels of cffDNA in women with early-onset pre-eclampsia (EOPE) and compared the results with those from healthy pregnancies. They found that the levels of cffDNA were significantly higher in women with EOPE, suggesting that cffDNA can be utilized as a biomarker for the early detection of EOPE [107].

Pre-term labor is considered a major cause of neonatal morbidity and mortality. It is defined as a labor that begins before the 37th week of pregnancy and is caused by uterine contractions. These contractions affect the uterine cervix with a risk of cervical ripening and membrane rupture [183]. It is not clear whether the elevations of cffDNA in women with preterm labors and delivery precede the clinical event and whether cffDNA can be used as a surrogate predictive marker. It is generally accepted that cffDNA is released due to the early initiation of the breakdown of the placental barrier [184].

### 2.4. Epigenetic Alterations in cffDNA

An invasion in surrounding tissues and rapid cell growth are both hallmarks of tumorigenesis and placentation, but in contrast to tumorigenesis, regulatory mechanisms are active and essential for normal placentation [185]. Tumor suppressor genes, such as *RASSF1A*, *APC*, *PRKCDBP*, and *P16*, are reported to be involved in the regulation of placentation and that their silencing is crucial for human cancer development [186,187,188,189,190].

Rahat et al. showed that the regulation of expression of the tumor suppressor genes *RASSF1A*, *APC*, and *PRKCDBP* by methylation is essential for normal placental development. They also found that hypermethylated promoter regions of these genes led to their downregulation, which was related to the development of gestation trophoblastic disease (GTD). In addition, decreased expression of *RASSF1A* and *P16* was observed in pregnancies complicated with PE [191]. Another study showed hypermethylated *RASSF1A* sequences of fetal origin in PE as well [192], thus demonstrating that these genes could potentially be used as fetal DNA epigenetic markers. 

As mentioned previously, *RASSF1A* gene methylation has been explored as a promising strategy to distinguish cffDNA from maternal cfDNA [149]. cffDNA was further explored in pregnancies complicated by IUGR, PE, and placental previa (PP), where the concentration of hypermethylated *RASSF1A* was significantly higher than in normal pregnancy at 15–28 weeks of gestation [193]. Kim et al. showed that the combination of the first-trimester cffDNA, cfDNA, and the epigenetic markers *DSCR3* (Down syndrome critical region gene 3), *RASSF1A*, and *HYP2* (translation elongation factor eIF-5A) with PAPP-A improved the predictive value for PE and hypertensive disorders of pregnancy (HDP) [194]. Another promising epigenetic biomarker for PE is *CMYC* (MYC proto-oncogene) [195]. Since the methylation level of *CMYC* promoter in maternal plasma mimics the methylation status of pre-eclamptic villi, *CMYC* could be a possible specific fetal epigenetic marker for PE [195]. 

Although the analysis of cffDNA is a promising non-invasive method to determine fetal RhD blood group type, fetal sex when diagnosing sex-linked disorders, fetal monogenic diseases, and fetal chromosomal aneuploidy, this type of NIPT is not an efficient method to diagnose common diseases of pregnancy [196]. Therefore, NIPT is considered a highly sensitive screening test and consequently, any positive result should be confirmed by an invasive testing [197,198,199], such as amniocentesis or karyotyping on cultured chorionic villi to avoid confined placental mosaicism. This is due to a strong evidence of discordant results (predominantly false positive, but also some false-negative) [200] between NIPT using cffDNA and conventional karyotyping following invasive procedures [201]. The discordant results can be attributed to confined placental mosaicism [202,203] and early demise of an aneuploid co-twin (vanishing twin) whose placenta continues to shed cffDNA [200]. In addition, it can be a result of maternal mosaicism [204] as sequencing yields both maternal and fetal cfDNA, and abnormal maternal tumor-derived cfDNA [205]. The failure of NIPT or inconclusive results, usually caused by a low fetal fraction of cffDNA, occur in up to 5% of cases [206]. Therefore, new fetal-specific DNA biomarkers should be identified to optimize the clinical utility of cffDNA.

## 3. Extracellular Vesicles

The term “extracellular vesicle” refers to four specific forms of cell-derived structures that differ, among other characteristics, by size: oncosomes (1–10 µm), apoptotic bodies (100–5000 nm), microvesicles (shedding particles; 100–1000 nm), and exosomes (40–100 nm) [28]. While these vesicles were initially considered a mechanism used by living or dying cells to reduce a waste material, they are specific because they are a part of a networking system between the cells [207]. EVs are secreted by various cell types and found in all body fluids. One of their most important roles is their ability to transfer genetic information to target cells and their powerful effect on the immune system [208].

The advantage of transferring information through vesicles is that it is independent of direct cell–cell contact and thus, it can transmit information to sites remote from the source cell [27]. Furthermore, proteins and nucleic acids are concentrated in the vesicles, and their structures and biological activities are preserved [209]. Taken together, there seems to exist a great potential for using EVs, especially exosomes, for both disease diagnosis and detection and the development of targeted therapies (Figure 1).

### 3.1. Exosomes

Although it has initially been thought that cells use exosomes solely for the removal of unwanted material, it is now clear that exosomes are involved in numerous other physiological processes and pathological states [210]. Exosomes originate from the endosomal cell compartment and differ from one another by the content they carry. This content is specific for their cells of origin. Thus, exosomes transmit valuable information about these cells and serve as indicators of their state [29]. As their formation is a combined result of endocytosis, intracellular processes, and exocytosis, exosomes represent the sum of these steps and are consequently composed of a wide variety of lipids, proteins, DNA molecules, RNA molecules, and metabolites of producing cells. Molecules found in the extracellular matrix, likely due to exocytosis, enter the neighboring cells by endocytosis, interact with their content, become processed into multivesicular bodies (MVB), and then exit the cells [211]. Exosomes facilitate the horizontal transfer of proteins, mRNAs, and miRNAs, thus inducing genetic and epigenetic changes of targeted cells [212]. 

Exosomes display a broad spectrum of homeostatic signaling mechanisms playing important roles in numerous physiological processes [213]. They are involved in the regulation of the immune system, blood coagulation, neuronal communication, sperm maturation, wound healing, and regulation of immune response against a fetus during pregnancy [214]. Aberrant signaling via exosomes could lead to various pathological conditions, such as metabolic diseases, infections, autoimmune diseases, inflammations, cardiovascular diseases, cancer, and pregnancy disorders [213,214,215].

We will further discuss the role of exosomes in placentas and cancers and their potential to serve as epigenetic biomarkers of pathological pregnancies and cancers. They may potentially be used for direct monitoring of fetal and placental status during pregnancy, i.e., for personalized gynecological protection through adjusted and timely treatment options. Exosomes carry products of syncytiotrophoblasts, such as proteins, lipids, glycans, and nucleic acids, which can signal to the mother the status of the placenta. Similarly, these particles may be useful in the diagnosis, treatment, and follow-up of cancers without the need for invasive procedures, such as tissue (needle) and surgical biopsies. They may serve as important biomarkers in the identification of various conditions, both normal and pathological. The greatest limitation for their use in clinical setting is the isolation process, which is most commonly performed using ultracentrifugal techniques [216]. Ultracentrifugation, however, may cause a contamination of exosomes with other protein aggregates or high-density lipoproteins, thus reducing the efficiency of their isolation. For this reason, size-exclusion chromatography has been increasingly used [217]. However, new techniques continue to be sought since there are still no standardized methods to isolate pure EV preparations. Other issues related to the collection of exosomes from human samples include factors such as physical activity undertaken before sampling and the time of day when the sample is collected that can affect the level of circulating exosomes [218]. In addition, some cells may produce vesicles in vitro after blood collection [219]. All these factors should be taken into account when choosing the best disease screening method.

#### 3.1.1. Exosomes in Cancer

Since exosomes play an important role in intercellular communications, they have been implicated in various human diseases, including cancer [220]. By acting on various recipient cells through transfer of oncoproteins and nucleic acids, exosomes modulate the activity of those cells and play crucial roles in cancer initiation, progression, and drug resistance [221]. The production and secretion of exosomes are increased in tumor cells in comparison with normal proliferating cells, so cancer patients have elevated levels of exosomes in their sera [222,223]. Tumor-derived exosomes are enriched in immunosuppressive proteins that have negative influence on antitumor immune responses, and they also carry tumor-associated antigens, co-stimulatory molecules, and major histocompatibility complex (MHC) components, which make them visible to the immune cells, thus stimulating an antitumor response [224]. This dual impact of tumor-derived exosomes on immune response presumably depends on the microenvironment, so the outcome depends on the composition of their cargo and the ability of recipient cells to accept or reject the transported signals [223].

#### 3.1.2. Exosomes as Epigenetic Cancer Biomarkers

Tumor-derived exosomes are also involved in epigenetic regulation. They contain proteins, DNA, or RNA that could induce epigenetic changes in recipient cells, including DNA, histone, miRNA, and lncRNA modifications. An impaired epigenome of recipient cells may initiate their neoplastic transformation.

Exosomal miRNAs are important players in epigenetic regulation of target cells. Unlike normal cells, tumor-derived exosomes contain various enzymes that are involved in miRNA synthesis and regulation, such as DICER, TRBP (TAR (trans-activation response) RNA-binding protein), and AGO2 (protein argonaute-2). These enzymes facilitate the transformation of the precursor miRNAs (pre-miRNAs) into mature miRNAs in a cell-independent manner. They also contain the membrane protein CD43, which regulates the accumulation of DICER [225]. Consequently, tumor-derived exosomes are regulators of efficient and rapid silencing of mRNAs, thus facilitating their oncogenic potential [225]. Exosomal miRNAs have some functional roles in cancer progression, such as receptor-mediated signaling and binding to toll-like receptors (TLRs) and transfer of phenotypic traits between cancer cells [226].

The miRNAs are the most abundant RNAs in exosomes [227]. They represent promising diagnostic, prognostic, and predictive cancer biomarkers (summary of their status in common cancers is provided in Table 2). Rabinowits et al. evaluated the circulating levels of 12 specific exosomal miRNAs previously linked to lung adenocarcinoma (miR-17-3p, miR-21, miR-106a, miR-146, miR-155, miR-191, miR-192, miR-203, miR-205, miR-210, miR-212, and miR-214) in patients with and without lung adenocarcinoma. They showed that exosomal miRNAs reflected the miRNA signature of the parental cancer, suggesting that these circulating exosomal miRNAs could be useful biomarkers for pulmonary adenocarcinoma [228]. In ovarian cancer, eight miRNAs (miR-21, miR-141, miR-200a, miR-200b, miR-200c, miR-203, miR-205, and miR-214) were found to be suitable diagnostic biomarkers for liquid biopsy profiling when isolated from tumor-derived exosomes from serum specimens [229]. Eichelser et al. showed that increased serum levels of circulating exosomal miR-373 were associated with triple negative and more aggressive breast carcinomas [230]. The levels of seven circulating exosomal miRNAs (let-7a, miR-1229, miR-1246, miR-150, miR-21, miR-223, and miR-23a) were elevated in primary CRC patients, even those with early-stage disease, and were significantly downregulated after surgical resection of tumors. The above-mentioned exosomal miRNAs are promising biomarkers for CRC [231]. In prostate cancer, four of the miRNAs analyzed by Endzeliņš et al. showed a diagnostic potential. The miR-200c-3p, miR-21-5p, and Let-7a-5p miRNAs showed a better diagnostic performance when tested in exosomes, while miR-375 performed better when analyzed in the whole plasma [232]. Exosomal miRNAs constitute only a small fraction of the plasma miRNA content [232,233,234], so this discrepancy between exosome-incorporated and whole plasma miRNA profiles, possibly due to specific sorting mechanisms for different miRNAs, indicates the importance of studying both the miRNAs for biomarker profiling.

Since exosomes and exosomal miRNAs are involved in cancer invasion and metastasis, they could serve as biomarkers of cancer cell invasion and metastasis. One of them is the miR-200 family. These miRNAs are involved in EMT and cell migration by targeting the E-cadherin transcriptional repressors *ZEB1/2* (zinc finger E-box binding homeobox 1 and 2) [264]. Le et al. showed that miR-200 family members use exosomes for being transferred to recipient cells, thus inducing the metastatic capability of breast cancer cells [236]. Another exosomal miRNA dysregulated in breast cancer is miR-10b. The exosome-mediated transfer of miR-10b promotes cell invasion in breast cancer, suppressing the protein level of its target genes, such as *HOXD10* (homeobox D10) and *KLF4* (Krüppel-like factor 4) [235]. In esophageal cancer, exosomal miR-21 promoted cell migration and invasion by targeting *PDCD4* (programmed cell death 4) and activating its downstream c-Jun N-terminal kinase (JNK) signaling pathway [238]. Sakha et al. found that exosomal miR-1246 induced cell motility and invasion by directly targeting tumor suppressor *DENND2D* (DENN/MADD domain-containing 2D) binding to its 3′-UTR in oral squamous cell carcinoma (OSCC) [240]. In glioblastoma, glioma-derived exosomal miR-148a promoted the proliferation and spread of glioblastoma cells via targeting *CADM1* (cell adhesion molecule 1), thus activating STAT3 pathway [242]. Exosomal miR-93 promoted the proliferation and invasion of hepatocellular carcinoma (HCC) by directly inhibiting *TP53INP1* (tumor protein p53-inducible nuclear protein 1), *TIMP2* (TIMP metallopeptidase inhibitor 2), and *CDKN1A* (cyclin-dependent kinase inhibitor 1A) [246]. Gong et al. showed that exosomal miR-675 from metastatic osteosarcoma promoted cell migration and invasion via *CALN1* (calneuron-1) [249].

For successful invasion, cancer cells have to communicate with the neighboring cells, possibly using exosomes as a signal transmission tool. To promote ovarian cancer cell invasion, exosomal miR-99a-5p increases the expression of the cell-adhesive glycoproteins, fibronectin and vitronectin, in neighboring peritoneal mesothelial cells [252]. Fang et al. showed that tumor-derived exosomal miR-1247-3p activated cancer-associated fibroblasts (CAFs) by targeting *B4GALT3* (β-1,4-galactosyltransferase 3), leading to activation of β1-integrin-NF-κB signaling in fibroblasts in HCC. Cancer progression was subsequently induced by CAFs, which secreted pro-inflammatory cytokines, including IL-6 and IL-8 [247]. Besides cytokines, CAFs could also regulate tumor progression via exosomes. A reduced expression of miR-34a-5p in CAF-derived exosomes contributes to OSCC cell proliferation and metastasis [265]. It was revealed that miR-34a-5p suppresses OSCC cell proliferation and metastasis by binding to its direct downstream target *AXL* [265]. Zang et al. showed that the loss of CAF-derived exosomal miR-320a contributes to HCC proliferation and metastasis [266]. They found that miR-320a directly targets *PBX3* (pre-B-cell leukemia transcription factor 3), thus suppressing the activation of the MEK/ERK MAPK pathway and, consequently, HCC cell proliferation [266]. Hence, the transfer of signals via exosomes could be reciprocal, from neighboring to cancer cells, and signals from the environment could affect the invasive potential of cancer cells as well.

Tumor-derived exosomes may also affect histone modification in the recipient cells. In CRC, tumor-derived miR-145 directly targets *HDAC11*, leading to IL-10 production by inhibition of histone deacetylation, thus polarizing macrophage-like cells into the M2-like phenotype. These M2-like tumor-associated macrophages promote CRC progression [267].

Besides miRNAs, exosomal lncRNAs may also reflect cancer growth, invasion, and metastasis (Table 3). The lncRNA MALAT1 (metastasis-associated lung adenocarcinoma transcript 1), found in EVs derived from NSCLC, has been implicated in cancer growth and migration [268]. In gastric cancer, exosomal lncRNA ZFAS1 (ZNFX1 antisense RNA 1) promoted the proliferation and migration of cancer cells [269]. SOX2OT (SOX2 overlapping transcript) lncRNA enriched in exosomes from pancreatic ductal adenocarcinoma cells, promotes EMT and stem cell-like properties by regulating *SOX2* (SRY-box 2) expression, thus inducing cancer invasion and metastasis [270]. Bian et al. found that lncRNA-ATB (lncRNA activated by TGF-β) could activate astrocytes by targeting miR-204-3p, thus promoting glioma cell invasion [271].

As mentioned previously, cfDNA is another nucleic acid that could be found in exosomes (Table 3). Exosomes could contain tumor-related DNA. Yoshida et al. found that the methylation status of *LINE1* (long interspersed element-1) and *SOX17* DNA in gastric juice-derived exosomes reflects the methylation status of nuclear DNA in gastric cancer. *SOX17* methylation was detected in both early and advanced gastric cancer, regardless of differentiation type, so it could be a useful biomarker in gastric cancer diagnosis [272]. An analysis of cfDNA is mostly based on the complete release of nucleic acids from bound proteins, lipids, and vesicles. Exosome isolation can be used to enrich ctDNA, but there should be no significant difference in the analysis of whole cfDNA in comparison with exosomal DNA, since more than 93% of amplifiable cfDNA in plasma is located in plasma exosomes.

#### 3.1.3. Placental Exosomes

Placenta communicates with other organs and tissues using EVs, among which exosomes seem to be the most relevant [273]. An important step forward was made when “placental fragments”, later called exosomes, were described in pregnant women [274]. Placental exosomes are formed by the same mechanism as exosomes of other cell types and tissues and originate from the endosomal partition of syncytiotrophoblasts [209]. The release kinetics and bioactivity of placental exosomes are under control of the placental micro-environment [275].

In contrast to other exosomes, placental exosomes lack MHC molecular expression. Instead, NK-cell receptor NKG2D-activating ligands (MICA/B, RAET1/ULBP1-5) are present on their surface [276,277]. They also contain pro-apoptotic proteins, such as Fas ligand (FasL), which is important for the development of immune tolerance, and tumor necrosis factor-related apoptosis-inducing ligand (TRAIL) [278]. These exosomes have an immunosuppressive effect on mothers during pregnancy, thus changing the maternal immune response important for pregnancy maintenance, and are important during lactation [220].

In early pregnancy, exosomes contribute to the process of embryo implantation and facilitate a communication between the embryo and endometrium [279]. They may be detected from gestational week 6 in the peripheral blood of pregnant women and reach the highest levels in the first trimester [280]. The ratio of placental to non-placental exosomes increases from mid-gestation, around week 20, and remains increased until birth [215]. The overall concentration of exosomes is significantly higher in pregnant than in non-pregnant women [281], in pathological pregnancies compared with normal pregnancies [282], and in pregnancies with term birth compared with those with pre-term birth [274]. PE, one of the most serious complications of pregnancy, is associated with an increase in both total and placental exosomes compared with physiological pregnancy, although this increase is specific only for the early onset of PE [280]. Furthermore, pregnant women with increased BMI have more exosomes in blood than those with normal weight, especially at later stages of gestation [215]. As mentioned above, the quantity of secreted exosomes, especially placental ones, differs between normal and pathological pregnancies, but the content of placental exosomes also differs between normal and pathological pregnancies [280].

Furthermore, exosomes have been detected in amniotic fluid. Given that they contain markers (CD24, anexin-1, and aquaporin-2) very similar to those found in exosomes in the urine of newborns, they most likely originate from the fetus (fetal kidneys) [209]. They could potentially be useful as indicators of the development of fetal kidneys and possibly other organs and as indicators of metabolic changes in the fetus [209].

A study of exosomes isolated from the peripheral blood of pregnant women, ex vivo placental transplants, or cultured trophoblasts has given us the most information about the exosomes [283,284]. They may serve as a form of placental “biopsy specimens” and provide information about the trophoblast condition and a broader picture than the classic biopsy, but in a non-invasive manner [216]. 

One of the specificities of placental exosomes is the presence of placental alkaline phosphatase (PLAP) in their content [281]. Alkaline phosphatase is not specific to placenta. It is found in other tissues as well, but unlike other alkaline phosphatases, PLAP lacks the last 24 amino acids, which makes it more specific [285]. This change provides PLAP with heat stability, resistance to chemical inactivation, and substrate specificity [285]. Its main functions described so far are assistance in the transfer of immunoglobulin G (IgG) from a mother to the fetus and stimulation of fibroblasts’ DNA synthesis and proliferation [285].

#### 3.1.4. Placental Exosomes as Epigenetic Biomarkers of Pathological Pregnancies

Another feature of syncytiotrophoblasts is a chromosome 19 miRNA cluster (C19MC) [286], whose concentration slowly decreases in mothers’ plasma after delivery. Placenta-specific miRNAs have a role in communication with mothers’ tissues, and placental exosomes are responsible for their release from syncytiotrophoblasts [280]. C19MC has 46 miRNAs [287]. These miRNAs control the migration and invasion of human trophoblasts and are involved in intercellular communication during pregnancy [288]. Takahashi et al. presented a novel model of exosomal placenta-associated miRNA functioning in cell–cell communication, where chorionic villous trophoblast exosomal C19MC miR-520c-3p regulates cell invasion by targeting *CD44* in extravillous trophoblasts [289]. An aberrant expression of C19MC miRNAs may affect the migration and invasion of trophoblasts. An increased expression of miR-517a/b and miR-517c contributes to decreased trophoblast invasion leading to PE [290]. The migration and invasion of trophoblasts can also be suppressed by miR-520g and miR-519d-3p via targeting *MMP2* (matrix metalloproteinase-2) that may contribute to the development of PE [291,292]. Apart from the above-mentioned miRNAs, PE is also associated with other C19MC miRNAs. MiR-519a was identified as a potential marker of severe PE [293], while miR-517-5p and miR-423-5p were proposed as potential biomarkers for the early stage of PE [294,295]. An aberrant expression of C19MC miRNAs could lead to other pregnancy-associated complications. The expression of four C19MC-derived mRNAs (miR-518b, miR-1323, miR-520h, and miR-519d) was decreased in IUGR placentas [296], while nine C19MC mRNAs (miR-515-5p, miR-516-5p, miR-518b, miR-518f-5p, miR-519a, miR-519e-5p, miR-520a-5p, miR-520h, and miR-526b-5p) were upregulated in women with pre-term birth [297].

Beside C19MC miRNAs, there are other miRNAs that are also involved in the migration and invasion of trophoblasts. The miRNA, MiR-155, inhibits the proliferation and migration of trophoblasts by downregulating cyclin D1 [298]. In PE, miR-210 was upregulated, which resulted in the inhibition of trophoblast invasion and migration [299]. A decreased expression of miR-34a in placenta accreta led to an increased trophoblast invasive potential [300]. Downregulated miR-195 in PE decreased trophoblast cell invasion via modulating *ACTRIIA* (activin A receptor type 2A) expression [301]. MiR-376c promoted trophoblast invasion and migration by targeting *ALK7* (activin receptor-like kinase 7) and *ALK5* (activin receptor-like kinase 5) to impair Nodal/TGF-β signaling [302], while miR-378a-5p induced trophoblast migration and invasion by inhibiting the expression of *NODAL* (nodal growth differentiation factor) [303].

A majority of the above-mentioned miRNAs are isolated from placental tissues by classic biopsy, but they could be candidate molecules to study exosomes non-invasively by using liquid biopsy. Cell-free miRNAs can circulate in the maternal bloodstream within EVs, called exosomes, which provide protection from degradation [304], or in a vesicle-free form associated with AGO2, a key component of the RNA-induced silencing complex (RISC) [305]. In most studies, whole plasma or serum was used as a source of circulating miRNAs, but it seems that exosomes are a better source of miRNA for biomarker profiling. Exosomes protect RNA molecules from degradation in the bloodstream, where large amounts of RNAses are present, and may be enriched with rare highly specific miRNAs of their cell of origin, providing a consistent source of miRNAs for disease biomarker detection [233]. This is in concordance with the study of Higashijima et al. who showed that the expression of four placenta-specific miRNAs was reduced in the placenta with IUGR, whereas circulating miRNA levels in maternal plasma showed no significant differences between uncomplicated and IUGR pregnancies [296]. It is important to study both exosomal miRNAs and cell-free miRNAs in whole plasma for biomarker profiling because of the discrepancy between exosome-incorporated and whole plasma miRNA profiles [232,233,234].

Although all functions and activities of placental exosomes are yet to be fully determined, it is certain that they provide a great potential as diagnostic and prognostic markers for a variety of pregnancy-related pathologies, such as PE and IUGR. Taken together, the current evidence indicates that the amount and content of placental exosomes along with total exosomes and gestational age could be used as a reliable marker for assessing the placenta, although additional studies are needed to confirm the usefulness of this approach in a clinical setting.

### 3.2. Other Extracellular Vesicles and Their Role in Placenta and Cancer

Other types of EVs include microvesicles, oncosomes, and apoptotic bodies. These vesicles are also involved in numerous cellular processes including cancer and diseases of pregnancy [28].

Microvesicles are important mediators of extracellular communication during cancer progression [306]. They help cancer cells to escape from immune surveillance and play vital roles in cancer invasion and metastases, inflammation, coagulation, and stem cell renewal and expansion [306,307]. Microvesicles could induce epigenetic changes by affecting DNA methylation status in the recipient cells [308]. Zhu et al. showed that microvesicles derived from leukemia cells attributed to the increased level of *DNMT3a* and *DNMT3b* mRNAs and their proteins in recipient cells via transmission of RNAs, leading to increased global DNA methylation levels [309]. Besides DNMT3a and DNMT3b, leukemia-derived microvesicles also increased the protein and mRNA levels of AICDA (activation-induced cytidine deaminase) in recipient cells. Taken together, these results show that malignant transformation from normal to leukemia cells may be induced by genomic instability promoted in recipient cells [309].

Microvesicles have also been shown to be involved in pathological pregnancies. Proteomic analysis of microvesicles from patients with PE revealed that microvesicle-derived proteins include those that are implicated in immune system response, clotting cascade, oxidative stress, apoptosis, and lipid metabolism pathways, suggesting their pathophysiological relevance in PE [310].

Oncosomes are large cancer-derived EVs associated with an advanced stage of the disease. Oncosomes are created by shedding of membrane blebs [311]. Both oncosomes and cancer-associated exosomes play important roles in the growth and spread of cancer cells, but their functions differ depending on their cellular source and protein content [311]. Their contents have not been studied in detail, but it seems that proteins present in cancer-associated exosomes are largely responsible for cell adhesion, cell mobility, and response to hypoxia, whereas those in oncosomes are involved in metabolic processes [311,312].

Apoptotic bodies are EVs released during apoptosis and represent the fragments of dying cells [30]. There are few studies that explored the role of apoptotic bodies and their effects on neighboring cells, and further research will be necessary to determine their roles in pathological pregnancies and cancer.

Since all these vesicles can contain mRNAs, miRNAs, and proteins that could induce epigenetic changes in recipient cells [308], they could also serve as epigenetic biomarkers in numerous diseases, including pathological pregnancies and cancer (Figure 1).

## 4. Conclusions

Liquid biopsy seems to be a novel, promising, and non-invasive tool for molecular profiling in precision medicine. Epigenetic changes in cfDNA (mostly DNA methylation) obtained by liquid biopsy are potential new biomarkers in pregnancy-related disorders and cancer. They could be useful not only for detection, but also for monitoring disease therapies. Exosomes, as well as other EVs, contain proteins, mRNAs, and miRNAs that can induce epigenetic changes in targeted cells, so they could also serve as epigenetic biomarkers for numerous diseases, including pathological pregnancies and cancer. Further studies are required to establish the clinical relevance of these biomarkers.

## Figures and Tables

**Figure 1 cells-08-01459-f001:**
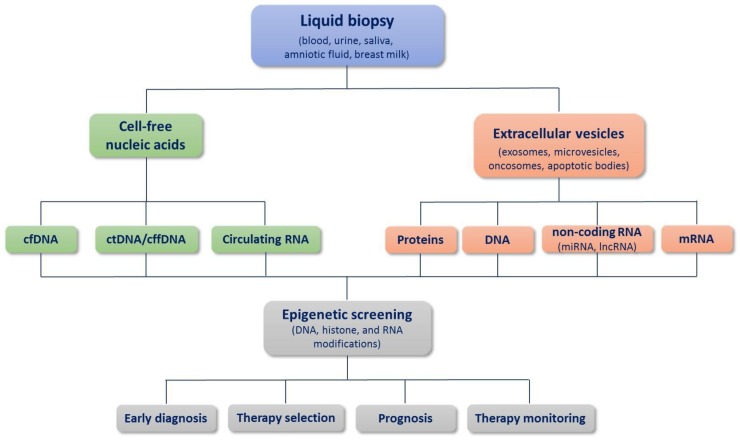
A diagram showing the potential utility of liquid biopsy highlighting cell-free nucleic acids and extracellular vesicles. These may undergo diverse epigenetic alterations that may have diagnostic, predictive, and prognostic values. cfDNA, cell-free DNA; ctDNA, cell-free tumor DNA; cffDNA, cell-free fetal DNA; miRNA, microRNA; lncRNA, long non-coding RNA.

**Table 1 cells-08-01459-t001:** The role of circulating cell-free tumor DNA (ctDNA) methylation in the detection of some common cancers.

Cancer	Biomarker	Findings	Refs.
Colorectal cancer (CRC)	*SEPT9*	The blood *SEPT9* gene methylation assay exhibited better performance in a symptomatic population than in an asymptomatic population when compared with other colorectal cancer screening tests.	[119]
		*SEPT9*-reinforced promoter DNA methylation in ctDNA was a significant discriminator between premalignant and malignant alterations in colorectal carcinogenesis.	[120]
	*WIF1*, *NYP*	DNA methylation of *WIF1* and *NPY* genes showed that hypermethylation of their gene promoters was detectable in both CRC tissue and corresponding ctDNA.	[121]
	*BCAT1*, *IKZF1*	Post-surgery detection of methylated *BCAT1* and *IKZF1* in ctDNA in CRC patients was related to increased risk of residual disease and subsequently recurrence.	[122]
Ovarian cancer	*RASSF1*, *BRCA1*	Tumor cell-specific hypermethylation of *BRCA1* and *RASSF1A* promoter regions were found in serum and peritoneal fluid from ovarian cancer patients.	[123]
Breast cancer	*WNT5A*, *KLK10*, *MSH2*, *SOX17*,	*WNT5A*, *KLK10*, *MSH2*, and *SOX17* genes were found more frequently methylated in ctDNA in all the patient groups than in healthy individuals. More intense methylation of *WNT5A* gene correlated with greater tumor size, advanced disease, and poor outcome; *SOX17* methylation was significantly correlated with the incidence of death, shorter progression-free survival, and overall survival, while *KLK10* methylation was significantly correlated with unfavorable clinicopathological characteristics and relapse.	[124]
	*APC*, *CCND2*, *FOXA1*, *PSAT1*, *RASSF1A*, *SCGB3A1*	Methylated *APC*, *CCND2*, *FOXA1*, *PSAT1*, *RASSF1A*, and *SCGB3A1* in ctDNA in breast cancer patients were confirmed as cancer-specific, although with variable performance. A panel combining four of them (*APC*, *FOXA1*, *RASSF1A*, and *SCGB3A1*) revealed the highest accuracy for disease detection (94%).	[125]
Lung cancer	*CDO1*, *HOXA9*, *AJAP1*, *PTGDR*, *UNCX*, *MARCH11*	Promoter methylation of a panel of six genes *(CDO1, HOXA9, AJAP1, PTGDR, UNCX,* and *MARCH11)* in ctDNA was found to have potential as a diagnostic and prognostic biomarker in non-small-cell lung cancer (NSCLC) patients.	[126]
	*CALCA*, *HOXA9*, *RASSF1A*, *CDKN2A*, *DLEC1*, *CDH13*, *PITX2*, *WT1*	Methylation levels of *CALCA*, *HOXA9*, *RASSF1A*, *CDKN2A*, *DLEC1*, *CDH13*, *PITX2*, and *WT1* in ctDNA in patients with lung cancer were significantly higher than in the non-lung cancer group. Three of them (RASSF1A, CDKN2A, and DLEC1) were found only in lung cancer patients.	[127]
	*B3GAT2*, *BCAR1*, *HLF*, *HOPX*, *HOXD11*, *MIR1203*, *MYL9*, *SLC9A3R2*, *SYT5*, *VTRNA1-3*	Using genome-wide plasma cfDNA methylation profiling, ten genes (*B3GAT2*, *BCAR1*, *HLF*, *HOPX*, *HOXD11*, *MIR1203*, *MYL9*, *SLC9A3R2*, *SYT5*, and *VTRNA1-3*) were identified as potential biomarkers for lung cancer.	[128]
Prostate cancer	*GSTP1*, *APC*	Epigenetic markers *GSTP1* and *APC* in circulating cell-free DNA have the potential to serve as prognostic markers for survival of castration-resistant prostate cancer patients.	[129]
Cholangiocarcinoma	*OPCML*, *HOXD9*	cfDNA methylation of *OPCML* and *HOXD9* may serve as a differential biomarker of cholangiocarcinoma and other biliary diseases.	[130]

SEPT9, septin-9; WIF1, Wnt inhibitory factor 1; NYP, neuropeptide Y; BCAT1, branched-chain amino acid transaminase 1; IKZF1, IKAROS family zinc finger 1; RASSF1, Ras association (RaIGDS7AF-6) domain family number 1; BRCA1, breast cancer 1; WNT5A, Wnt family member 5A; KLK10, kallikrein-10; MSH2, mutS protein homolog 2; SOX17, SRY-box 17; APC, adenomatous polyposis coli; CCND2, cyclin-D2; FOXA1, forkhead box A1; PSAT1, phosphoserine aminotransferase 1; SCGB3A1, secretoglobin family 3A member 1; CDO1, cysteine dioxygenase type 1; HOXA9, homeobox A9; AJAP1, adherens junction-associated protein 1; PTGDR, prostaglandin D2 receptor; UNCX, UNC homeobox; MARCH11, membrane-associated ring-CH-type finger 11; CALCA, calcitonin-related polypeptide alpha; CDKN2A, cyclin-dependent kinase inhibitor 2A; DLEC1, DLEC1 cilia and flagella associated protein; CDH13, cadherin-13; PITX2, paired-like homeodomain 2; WT1, Wilms’ tumor suppressor gene; B3GAT2, beta-1,3-glucuronyltransferase 2; BCAR1, BCAR1 scaffold protein, Cas family member; HLF, hepatic leukemia factor; HOPX, HOP homeobox; HOXD11, homeobox D10; MIR1203, miR-1203; MYL9, myosin light chain 9; SLC9A3R2, SLC9A3 regulator 2; SYT5, synaptotagmin-5; VTRNA1-3, vault RNA 1-3; GSTP1, glutathione S-transferase pi 1; OPCML, opioid-binding protein/cell adhesion molecule-like; HOXD9, homeobox D9.

**Table 2 cells-08-01459-t002:** Exosomal microRNAs (miRNAs) and their targets that are involved in cancer development.

Cancer	miRNAs	Target	Refs.
Breast cancer	miR-10b	*HOXD10*, *KLF4*	[235]
	miR-373	Unknown	[230]
	miR-200 family members	Unknown	[236]
	miR-1246	Unknown	[237]
Esophageal cancer	miR-21	*PDCD4*	[238]
	miR-93-5p	*PTEN*	[239]
Oral squamous cell carcinoma	miR-1246	*DENND2D*	[240]
Glioblastoma	miR-301a	Unknown	[241]
	miR-148a	*CADM1*	[242]
Hepatocellular carcinoma	miR-665	Unknown	[243]
	miR-125b	Unknown	[244]
	miR-210	*SMAD4*, *STAT6*	[245]
	miR-93	*TP53INP1*, *TIMP2*, *CDKN1A*	[246]
	miR-1247-3p	*B4GALT3*	[247]
	miR-103	Unknown	[248]
Osteosarcoma	miR-675	*CALN1*	[249]
Lung cancer	miR-17-3p, miR-21, miR-106a, miR-146, miR-155, miR-191, miR-192, miR-203, miR-205, miR-210, miR-212, miR-214	Unknown	[228]
	miR-21-5p, miR-126-3p, miR-140-5p	Unknown	[250]
	miR-193a-3p, miR-210-3p, miR-5100	Unknown	[251]
Ovarian cancer	miR-21, miR-141, miR-200a, miR-200b, miR-200c, miR-203, miR-205, miR-214	Unknown	[229]
	miR-99a-5p	Unknown	[252]
	miR-375, miR-1307	Unknown	[253]
Colorectal cancer	let-7a, miR-1229, miR-1246, miR-150, miR-21, miR-223, miR-23a	Unknown	[231]
	miR-6803-5p	Unknown	[254]
	miR-27a, miR-130a	Unknown	[255]
	miR-25-3p	*KLF2*, *KLF4*	[256]
Prostate cancer	miR-200c-3p, miR-21-5p, let-7a-5p	Unknown	[232]
Pancreatic cancer	miR-23b-3p	Unknown	[257]
	miR-191, miR-21, miR-451a	Unknown	[258]
Gastric cancer	miR-423-5p	*SUFU*	[259]
	miR-23b	Unknown	[260]
	miR-130a	*C-MYB*	[261]
	miR-27a	Unknown	[262]
Acute myeloid leukemia	miR-125b	Unknown	[263]

HOXD10, homeobox D10; KLF4, Krüppel-like factor 4; PDCD4, programmed cell death 4; PTEN, phosphatase and tensin homolog; DENND2D, DENN/MADD domain-containing 2D; CADM1, cell adhesion molecule 1; SMAD4, mothers against decapentaplegic homolog 4; STAT6, signal transducer and activator of transcription 6; TP53INP1, tumor protein p53-inducible nuclear protein 1; TIMP2, TIMP metallopeptidase inhibitor 2; CDKN1A, cyclin-dependent kinase inhibitor 1A; CALN1, calneuron-1; B4GALT3, β-1,4-galactosyltransferase 3; KLF2, Krüppel-like factor 2; SUFU, suppressor of fused homolog; C-MYB, v-myb avian myeloblastosis viral oncogene homolog.

**Table 3 cells-08-01459-t003:** Exosomal long non-coding RNAs (lncRNAs) and exosomal cell-free DNA (cfDNA) involved in the development of some common cancers.

	Biomarker	Cancer	Refs.
**lncRNA**	MALAT1	Non-small-cell lung cancer	[268]
	ZFAS1	Gastric cancer	[269]
	SOX2OT	Pancreatic ductal adenocarcinoma	[270]
	lncRNA-ATB	Glioma	[271]
**cfDNA**	*LINE1* and *SOX17* methylation	Gastric cancer	[272]

MALAT1, metastasis-associated lung adenocarcinoma transcript 1; ZFAS1, ZNFX1 antisense RNA 1; SOX2OT, SOX2 overlapping transcript; lncRNA-ATB, lncRNA activated by TGF-β; LINE1, long interspersed element-1; SOX17, SRY-box 17.

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
