# Peer review of "Novel Epigenetic Biomarkers in Pregnancy-Related Disorders and Cancers"

_cells, 2019, doi:10.3390/cells8111459_

Round 1

Reviewer 1 Report

This is a comprehensive and up-to-date review on extracellular vesicles and circulating nucleic acids that can potentially be used as epigenetic biomarkers to diagnose or follow-up diseases like cancer and disturbances of pregnancy. The manuscript reads well and cites and discusses the most relevant literature. However, before it can be formally accepted for publication in "Cells", the authors should rectify a few issues (mostly phrasings, typos and abbreviations):

In the Introduction, all the structures and nucleic acids shown in Figure 1 should be briefly mentioned in the text, especially miRNAs, i.e. in the first paragraph of page 4.

The tables occupy too much space. In particular, Table 2 should be condensed.

Lines 112-115 contain many references that have not been numbered.

Line 163: Please add "of" between "time" and "tissue".

Line 171: "responds" must be replaced by "corresponds".

Line 172: The term "nucleus-" can be removed.

Line 186: ...been suggested to play active roles...

Line 274: The sentence "..does not necessarily mean the cancer diagnosis" is unclear and must be rephrased.

Line 280: Please introduce the sentence with "The".

Line 294: Add "the" between "that" and "sensitive".

Line 342 and 392: If the gene name is used in capitals and italics (i.e. SEPT9) then the term "gene" behind it may be omitted.

Lines 344 and 390: typo: "promotOr".

Line 353: Please replace "related" with "associated".

Line 378: The abbreviation for APC should be introduced earlier at first mention (line 361).

Line 391: change "genes" into "gene".

Line 395: remove "the" in front of "important".

Line 397: move "the" from "affected patients" to "above-mentioned". In general, the usage of definite articles is not always correct. Sometimes one is missing when its required and vice versa. Please check manuscript.

Line 423: ..singular: is the villous trophoblast, and remove "the" in front of "pregnancy".

Line 439: ..have shown.. Line 601: remove "environment".

Line 612: The introduction of "miRNA" for "microRNA" should be done earlier.

Line 672: The sentence extending from line 672 to line 674 is redundant and should be rephrased and shortened.

Line 685: change "target" to "targets". Specify the type of MAPK pathway (MEK/ERK or p38?).

Line 690 and 698: change "targeted" to "targets", and Promoted" to "promotes". In general, please be consistent in using either past or present tense when describing the results from other studies.

Line 801: please rephrase sentence as follows: ...but it seems that exosomes are a better source of ..or similar.

Line 812: please insert "a" between "for" and "variety".

Line 815: change "assess" to "for assessing" or "in order to assess".

Author Response

This is a comprehensive and up-to-date review on extracellular vesicles and circulating nucleic acids that can potentially be used as epigenetic biomarkers to diagnose or follow-up diseases like cancer and disturbances of pregnancy. The manuscript reads well and cites and discusses the most relevant literature. However, before it can be formally accepted for publication in "Cells", the authors should rectify a few issues (mostly phrasings, typos and abbreviations):

In the Introduction, all the structures and nucleic acids shown in Figure 1 should be briefly mentioned in the text, especially miRNAs, i.e. in the first paragraph of page 4.

Thanks for your comment. This suggestion is added (please see page 3, lines 96-104 and page 5, lines 111-123).

The tables occupy too much space. In particular, Table 2 should be condensed.

Fully agree. The tables are now condensed.

Lines 112-115 contain many references that have not been numbered.

Thank you for this comment. The references are adjusted according to the MDPI style (please see page 5, line 144).

Line 163: Please add "of" between "time" and "tissue".

This correction is added (please see page 6, line 199).

Line 171: "responds" must be replaced by "corresponds".

This correction is added (please see page 6, line 208).

Line 172: The term "nucleus-" can be removed.

This correction is added (please see page 6, line 208).

Line 186: ...been suggested to play active roles...

This correction is added (please see page 7, lines 224-225).

Line 274: The sentence "..does not necessarily mean the cancer diagnosis" is unclear and must be rephrased.

Thank you for your suggestion. The following changes have been made: page 9, lines 318-320.

Line 280: Please introduce the sentence with "The".

This correction is added (please see page 9, lines 326).

Line 294: Add "the" between "that" and "sensitive".

We appreciate your suggestion, but we think that in this case word ‘is’ is better option than word ‘the’ (please see page 9, line 342).

Line 342 and 392: If the gene name is used in capitals and italics (i.e. SEPT9) then the term "gene" behind it may be omitted.

This suggestion is added (please see pages 14-15, lines 397-451).

Lines 344 and 390: typo: "promotOr".

These corrections are added (please see page 14, line 398 and page 15, line 449).

Line 353: Please replace "related" with "associated".

This correction is added (please see page 14, line 409).

Line 378: The abbreviation for APC should be introduced earlier at first mention (line 361).

We fully agree that we should introduce the abbreviation at first mention, but beside APC, there are many other genes listed in parentheses in this paragraph (please see pages 14-15, lines 419-455). We think that it will be a little bit chaotic if we introduce full names and abbreviation for each listed gene. If you think that it is necessary, we will accept your suggestion and add corrections.

Line 391: change "genes" into "gene".

This correction is added (please see page 15, line 451).

Line 395: remove "the" in front of "important".

This correction is added (please see page 15, line 454).

Line 397: move "the" from "affected patients" to "above-mentioned". In general, the usage of definite articles is not always correct. Sometimes one is missing when its required and vice versa. Please check manuscript.

This correction is added (please see page 15, lines 456-457). Professional language editor has reviewed manuscript and all corrections have been made.

Line 423: ..singular: is the villous trophoblast, and remove "the" in front of "pregnancy".

These corrections are added (please see page 15, line 485).

Line 439: ..have shown.. Line 601: remove "environment".

These corrections are added (please see page 16, line 502 and page 19, line 675).

Line 612: The introduction of "miRNA" for "microRNA" should be done earlier.

These corrections are added (please see page 3, line 97 and page 19, line 687).

Line 672: The sentence extending from line 672 to line 674 is redundant and should be rephrased and shortened.

Thank you for your suggestion. We have rephrased and shortened this sentence (please see page 22, lines 748-749).

Line 690 and 698: change "targeted" to "targets", and Promoted" to "promotes". In general, please be consistent in using either past or present tense when describing the results from other studies.

Thank you for your valuable comment. The following changes have been made: page 22, line 767 and page 23, line 775.

Line 801: please rephrase sentence as follows: ...but it seems that exosomes are a better source of ..or similar.

This correction is added (please see page 25, line 883).

Line 812: please insert "a" between "for" and "variety".

This correction is added (please see page 25, line 894).

Line 815: change "assess" to "for assessing" or "in order to assess".

This correction is added (please see page 25, line 897).

Reviewer 2 Report

This is an overall interesting summary of the current literature on the analysis of liquid biomarkers for cancer and pregancy.

The one big issue with this manuscript is languague and writing style.

There are a great many imprecisions both in language and also partly in content, starting with a difficult to understand title over several issues in the abstract, (e.g. it is not really precision medicine that develops new drugs, the authors use the word "probe" wrong, etc.) into the introduction (Sigmund Freud, not really!), and on it goes.

I recommend professional language editing and also a second look over some conceptual decisions and then resubmission to deal with then remaining issues

Author Response

This is an overall interesting summary of the current literature on the analysis of liquid biomarkers for cancer and pregancy.

The one big issue with this manuscript is languague and writing style.

There are a great many imprecisions both in language and also partly in content, starting with a difficult to understand title over several issues in the abstract, (e.g. it is not really precision medicine that develops new drugs, the authors use the word "probe" wrong, etc.) into the introduction (Sigmund Freud, not really!), and on it goes.

Appreciate your comments and suggestions. The title has been changed to ‘Novel epigenetic biomarkers in pregnancy-related disorders and cancers’. Regarding other issues, the following changes have been made: pages 1, lines 21-36; page 2, lines 41-44 and line 83).

I recommend professional language editing and also a second look over some conceptual decisions and then resubmission to deal with then remaining issues.

Thanks for this important issue. Manuscript has been comprehensively reviewed by a professional English language editor and all corrections have been made. All the made changes are labeled with “track changes” option.